# Impact of Metal-Containing Industrial Effluents on Leafy Vegetables and Associated Human Health Risk

**DOI:** 10.3390/foods13213420

**Published:** 2024-10-27

**Authors:** Alexandra Kravtsova, Inga Zinicovscaia, Alexandra Peshkova, Pavel Nekhoroshkov, Liliana Cepoi, Tatiana Chiriac, Ludmila Rudi

**Affiliations:** 1Joint Institute for Nuclear Research, 6 Joliot-Curie Str., 1419890 Dubna, Russia; zinikovskaia@mail.ru (I.Z.); peshkova@jinr.ru (A.P.); nekhoroshkov@jinr.ru (P.N.); 2Department of Nuclear Physics, Horia Hulubei National Institute for R&D in Physics and Nuclear Engineering, 30 Reactorului Str. MG-6, 077125 Magurele, Romania; 3Doctoral School of Natural Sciences, Moldova State University, M. Kogalniceanu Str., 75A, MD-2009 Chisinau, Moldova; 4Institute of Microbiology and Biotechnology, Technical University of Moldova, 1, Academiei Str., MD-2028 Chisinau, Moldova; liliana.cepoi@imb.utm.md (L.C.); tatiana.chiriac@imb.utm.md (T.C.); ludmila.rudi@imb.utm.md (L.R.)

**Keywords:** antioxidant activity, biochemical parameters, bioconcentration factor, chromium, estimated daily intake, target hazard quotient, translocation factor, zinc

## Abstract

One of the primary sources of trace elements in the environment is wastewater used for irrigation. However, the effects of untreated wastewater containing high concentrations of chromium and zinc on vegetables and the potential human health risks associated with their consumption are poorly understood. This pot experiment aimed to address this research gap. The accumulation of chromium and zinc and their effect on the biochemical parameters of lettuce (*Lactuca sativa*) and green onion (*Allium fistulosum* L.) irrigated with untreated industrial effluents were assessed. The average concentrations of chromium and zinc in the edible parts of the vegetables ranged between 7.36 and 7.58 mg/kg dry weight and 59.8 and 833 mg/kg dry weight, respectively. The irrigation of the lettuce with the effluent containing zinc at a concentration of 2.95 mg/L led to a significant increase in the content of phenols and the antioxidant activity. A significant reduction in the chlorophyll content of the lettuce leaves and the antioxidant activity of the onion leaves was observed when the plants were irrigated with the effluent containing zinc at a concentration of 78 mg/L. No non-carcinogenic health risk from the intake of chromium and zinc was identified through the consumption of lettuce and green onion, primarily due to the fact that a smaller proportion of the total metal content was transferred to their edible parts.

## 1. Introduction

Wastewater treatment and reuse for crop production is a good example of wastewater recycling in developed countries [1,2,3]. However, in India, Pakistan and other developing countries, effluents from various industries and domestic wastewater are often discharged into surface water and land without any secondary or tertiary treatment [1,3,4]. Therefore, one of the main sources of trace metals, including toxic ones, in the environment is wastewater used for irrigation [2,3,5]. Continuous long-term irrigation with wastewater results in trace and toxic elements accumulating in soils, which leads to their further accumulation in food crops and can cause various chronic diseases through intake of contaminated vegetables and other crops [6,7].

According to the Food and Agriculture Organization (FAO), the harvested area for the main leafy vegetables has increased by more than 200% over the past decades, also due to the growing number of health-conscious consumers [8]. The WHO/FAO have recommended a daily intake of at least 400 g of fruit and vegetables/five servings per day (except potatoes and other starchy tubers). Green leafy vegetables were recommended to be one of those five servings [9]. Lettuce and green onions are popular and grown in many regions of the world [10,11]. Thus, the results of a survey on the consumption of leafy greens in São Paulo, Brazil, showed that lettuce was one of the most commonly consumed types of leafy vegetables [12]. Apart from the variety which green leafy vegetables add to the meal, they are valuable sources of minerals, phenolic compounds, ascorbic acid, vitamins A and K, folates, dietary fiber, proteins, carbohydrates and carotenoids [9,10,13]. The results of a prospective study of 960 participants aged 58–99 years in the USA also showed that consuming green leafy vegetables may help to slow down the decline in cognitive abilities with age, possibly due to the neuroprotective effects of lutein, folic acid, β-carotene and phylloquinone [14]. Leafy vegetables, known as a good source of K, Ca, Mg, P, and Fe, also contain such trace metals as Cu, Mn, Zn, Cr and others [9].

Chromium (III) and zinc are essential elements for humans [15,16,17]. They play an important role in the metabolism of glucose (chromium (III)), as well as in the processes of synthesis and degradation of proteins, carbohydrates, fats and nucleic acids, growth, division and cell differentiation (zinc) [15,16,17]. At the same time, a high concentration of zinc can cause impairment of growth and reproduction [18], and the hexavalent form of chromium (chromium (VI)), being toxic to humans, can cause DNA damage, hypoglycemia and diseases of the liver and kidneys [15,16].

Industrial effluent discharge is a significant anthropogenic source of zinc and chromium in the environment [16,17]. Zinc is used in the production of dry cell batteries and alloys, as well as in galvanization [16]. Widespread use of chromium in such industries as metallurgy and metalworking, electroplating, paints and dyes, timber processing, paper production, textiles and others has led to its release in large quantities in the environment [19,20].

It is known that leafy vegetables are able to accumulate a higher proportion of chromium and zinc in the vegetative parts compared to cereals or tubers [1]. Therefore, many studies have been devoted to the assessment of the risks to human health associated with the consumption of vegetables (including leafy ones) irrigated with treated wastewater [3,21,22,23]. However, there are very few studies on the effect of high concentrations of chromium and zinc in untreated wastewater on leafy vegetables and the potential human health risk associated with their consumption [3,24,25]. Moreover, published studies often do not report the metal concentrations in the effluents or chromium oxidation states, making it difficult to assess the wastewater toxicity.

Therefore, this study aims to address this research gap and evaluate the impact of high concentrations of chromium and zinc in irrigation wastewater on lettuce and green onions and the potential health risks associated with their consumption. To achieve this goal, the following tasks were set: (1) to assess the accumulation of chromium and zinc in soil and different parts of lettuce and green onion grown on soils irrigated with industrial effluents; (2) to measure the bioaccumulation capacity of lettuce and onion; (3) to estimate the chromium and zinc translocation in different parts of vegetables; (4) to calculate the estimated daily intake, target hazard quotient and hazard index of chromium and zinc through intake of wastewater-irrigated lettuce and onion; and (5) to determine the biochemical parameters and antioxidant activity of vegetables irrigated with industrial effluents.

## 2. Materials and Methods

### 2.1. Effluents

Two industrial effluents containing zinc in a concentration of 2.95 mg/L (E1) and 78.0 mg/L (E2), and effluent containing both zinc and chromium (VI) in concentrations of 6.85 mg/L and 9.96 mg/L, respectively (E3), were obtained from the electroplating units of the Scientific Production Association “Atom” (Dubna, Russia). The content of other metals (Fe, Mo, Sr, Ni, Mn, Co, Pb, Cd, Cu) in these effluents was below the detection limit or was significantly less than the maximum permissible concentrations established for these metals in irrigation water, which allowed us to ignore their influence when assessing the results. The mean pH values of E1, E2 and E3 were 6.85, 6.93 and 6.82, respectively.

### 2.2. Laboratory Experiment

The pot experiment was chosen because it allows a more accurate comparison of the effects of different metal concentrations on plants under the same conditions compared to the field studies. The soil and *Lactuca sativa* and *Allium fistulosum* L. seeds were ordered from the Buisky chemical plant and “Agroseedstrade” company, respectively (Russia). The soil had the following characteristics: pH—6.5, humidity ≤ 65%, NH_4_ + NO_3_—135 (mg/L), P_2_O_5_—180 (mg/L), K_2_O—315 (mg/L), Fe—0.8 (mg/kg), Mo—0.1 (mg/kg), B—0.4 (mg/kg), Zn—0.2 (mg/kg), Mn—8.0 (mg/kg), Cu—3 (mg/kg). The seeds were planted in plastic pots containing 1.0 kg of soil. The seeds were sown in May, which coincided with the normal growing season for these leafy vegetables. The plants were grown under conditions that were close to natural, with a photoperiod of 15 h of light/9 h of darkness, 23 ± 2 °C and relative humidity of about 55%.

The pots were watered three times per week with industrial effluents, whereas tap water was used as a control. The experiment lasted for six weeks. The total amount of water added to each pot during experiment was about 1500 mL. At the end of the experiment, the harvested vegetable samples were carefully rinsed in distilled and deionized water to remove soil particles from the surface, and separated into leaves (edible part) and roots (non-edible part). After being cleaned with deionized water, the fresh weight (FW) of the edible part of the vegetables in each pot was recorded. The samples of soil were taken from a depth of 0–10 cm. For the experiment, 8 plants of each species were analyzed for each type of effluent. The experiments were performed in three repetitions.

### 2.3. Sample Preparation and Data Analysis

#### 2.3.1. Chromium and Zinc Concentrations

All the samples were dried in an oven at 40 °C for 2–3 days, ground in a ball mill, and dried again in an oven at 105 °C for 1 day. Plant and soil samples weighing about 0.1 g and 0.2 g, respectively, were placed in standard Teflon vessels with a volume of 110 mL, where 3 mL of HNO_3_ (Honeywell Fluka 69% Suprapur, Sigma-Aldrich, Darmstadt, Germany) and 1 mL of H_2_O_2_ (Sigma-Aldrich 30% EMSURE, Darmstadt, Germany) were added for plant samples and 5 mL HNO_3_, 2 mL H_2_O_2_ and 1 mL HF (Merck 40% EMSURE, Darmstadt, Germany) for soil samples. Then, the vessels were left for 30 min at room temperature to remove any volatile compounds. The samples were digested using a MARS 6 microwave system (CEM, USA). The temperature rise time was 20 min, then the temperature was maintained at 180 °C for 15 min, and finally the vessels were cooled for 20 min. All the stages took place at a power of 290–1800 W; the power adjustment was performed automatically. Then, the samples were transferred into 10 mL and 50 mL flasks for the vegetation and soil samples, respectively, and made up to the mark with deionized water.

The chromium and zinc concentrations in the tap water, effluents, lettuce, onion and soils were determined by the inductively coupled plasma–optical emission spectrometry (ICP-OES) technique on the PlasmaQuant^®^ PQ9000 Elite spectrometer (Analytik Jena, Jena, Germany). High-purity argon was used as the plasma-forming gas. The measurements were applied in axial viewing mode. The signal measurements for each element were carried out three times. The signal reading time was 3 s. The chromium 205.552 nm and zinc 206.200 nm emission lines were chosen to calculate the chromium and zinc concentration, respectively, in the samples. The limit of detection (LOD) and the limit of quantification (LOQ) for zinc were 0.0059 mg/L and 0.0235 mg/L, respectively, and the LOD and LOQ for chromium were 0.0004 mg/L and 0.0016 mg/L, respectively.

For the ICP-OES calibration, IV-STOCK-13 (Inorganic Ventures, Christiansburg, VA, USA) was used, diluted in vessels with 2% HNO_3_ with a concentration of 0.001, 0.01, 0.1 and 1 mg/L. Deionized water was provided by the Adrona water purification system. As the zero value, 2% HNO_3_ (Honeywell Fluka 69% Suprapur, Sigma-Aldrich, Darmstadt, Germany) was used.

The following certified reference materials were used for quality control assurance: INCT-OBTL-5 (Oriental Basma Tobacco Leaves) and NIST 2709a (San Joaquin Soil Baseline Trace Element Concentrations). The recovery rate for chromium and zinc varied from 84 to 105% (Table 1).

#### 2.3.2. Pigment Content

Chlorophyll and carotenoids were extracted from the homogenized leaves’ biomass using 80% acetone. The extraction took place at room temperature under continuous stirring for 12 h. The resulting suspension was centrifuged for 5 min at 4000× *g*.

The absorbance (Abs) of the extracts was measured at 470, 645 and 663 nm. The absorbance was measured using a PG Instruments T80 UV-VIS spectrophotometer (UK), with acetone used as the blank. The content of chlorophyll and carotene was calculated by applying Equations (1)–(3) [26]:(1)$$Chlorophyll\;a={Abs}_{663}\times 12.21-{Abs}_{645}\times 2.81,$$
(2)$$Chlorophyll\;b={Abs}_{645}\times 20.13-{Abs}_{645}\times 5.03,$$
(3)$$Carotene=1000\times {Abs}_{470}-Chlorophyll\;a\times 3.27-Chlorophyll\;b\times 104/229.$$

#### 2.3.3. Antioxidant Activity and Phenols Content

To prepare the hydro-ethanolic extract, 1 g of dry biological material (onion leaves or lettuce leaves) was mixed with 10 mL of 50% ethyl alcohol solution. The obtained samples were subjected to maceration for 12 h under continuous stirring conditions. The obtained extracts were firstly separated from the biological material by centrifugation and then they were filtered and kept in the refrigerator at 0 °C.

Determination of the antioxidant activity by applying ABTS radical (2,2′-Azino-di-(3-ethylbenzthiazoline sulfonic acid) [27].

To determine the antioxidant activity, a reactant mixture consisting of 2.7 mL of ABTS working solution with an absorbance value of 0.700 ± 0.02 (mixture of 1:1 (*v*/*v*) 7 mM ABTS solution and 2.45 mM potassium persulfate solution, the duration of radical formation 12–16 h in the dark) and 0.3 mL of hydro-ethanolic extract from the plant material (onion leaves/lettuce leaves) was prepared. After 6 min of reaction, the absorbance at 734 nm was recorded. The value of the antioxidant activity was calculated using Formula (4) and expressed as the % of inhibition:(4)$$\%\;of\;inhibition=100-{Abs}_{Sample\;\;}\times 100/{Abs}_{ABTS},$$
where Abs*_sample_* is the absorbance of the sample at the end of the reaction time and Abs_ABTS_ is the absorbance of the ABTS working solution.

Determination of the antioxidant activity by applying DPPH radical (2,2-diphenyl-1-picrylhydrazyl) [28].

The reaction mixture consisted of 2.7 mL of DPPH solution (60 µM) and 0.3 mL of hydro-ethanolic extract from the plant material. The samples were incubated in the dark for 30 min at room temperature and then the absorbance at 517 nm was measured. The value of the antioxidant activity was expressed as the % of inhibition and calculated using Equation (5):(5)$$\%\;of\;inhibition=100-{Abs}_{Sample\;\;}\times 100/{Abs}_{DPPH},$$
where Abs*_sample_* is the absorbance of the sample at the end of the reaction time and Abs_DPPH_ is the absorbance of the DPPH working solution.

Determination of phenols content [29].

First, 0.3 mL of hydro-ethanolic extract of plant material was mixed with 1.5 mL of Folin–Ciocalteu reagent with a dilution of 1:9 and 1.2 mL of 7.5% sodium bicarbonate solution. The samples were subjected to incubation at 50 °C for 5 min. At the end of the incubation time, the absorbance of the samples was measured at 760 nm. Gallic acid solution was used as a standard. The content of phenols was expressed in mg gallic acid/100 g plant biomass.

#### 2.3.4. Bioconcentration and Translocation Factors

The bioconcentration factor (BCF) was calculated by dividing the concentration of the element in the edible part of the vegetable (C_L_, mg/kg dw) by the corresponding concentration in soil (C_s_, mg/kg dw) [30].
(6)$$\mathrm{BCF}=C_{L}/C_{S}$$

The transfer factor (TF) was used to evaluate the ability of leafy vegetables to translocate metals from roots (C_R_, mg/kg dw) to leaves (C_L_, mg/kg dw) [23].
(7)$$\mathrm{TF}=C_{L}/C_{R}$$

#### 2.3.5. Estimated Daily Intake

The estimated daily intake (EDI, mg/kg bw/day) of chromium and zinc via he consumption of wastewater-irrigated lettuce and onion was determined based on the metal concentration in the leaves of the vegetables (C_L_, mg/kg dry weight), daily intake of leafy vegetables (D_IV_, kg per day), conversion factor (C_F_) and average body weight (ABW, kg), and it was calculated as follows [31]:(8)$$\mathrm{EDI}=C_{L}\;\times \;D_{IV}\times C_{F}/\mathrm{ABW}.$$

The conversion factors of 0.054 for lettuce and 0.049 for green onion, as calculated in this experiment, were used to convert the dry vegetable weight to the fresh green vegetable weight.
(9)Convertion factor=(100−% of moisture)/100

To calculate the % of moisture in the edible part of the lettuce and green onion, the fresh and dry leaves of the vegetables were weighted during the experiment.

The average body weight of an adult was considered to be 70 kg. Consumption data were estimated by accessing surveys for different regions of the world [32,33]. The per capita intake of leafy vegetables ranged from 37 to 43 g/day per an adult, weighted as 70 kg. Based on the literature data, the daily intake of 0.04 kg/day of leafy vegetable was used in this study.

#### 2.3.6. Target Hazard Quotient

The target hazard quotient (THQ), developed by the USEPA (United States Environmental Protection Agency), fits well for the purpose of non-carcinogenic health risk assessment. If the THQ is <1, then no non-carcinogenic health effects are expected. If the THQ is >1, then there is a possibility of adverse health effects. The THQ was calculated using the following equation [34]:(10)$$THQ=E_{F}\times E_{D\;}\times EDI/RfD\times AT\;\;$$
where *E_F_* is the exposure frequency (365 days/year); *E_D_* is the exposure duration (70 years); *EDI* is the estimated daily intake (mg/kg bw day); *RfD* is the oral reference dose (0.003 mg/kg day^−1^ for chromium and 0.3 mg/kg day^−1^ for zinc); and *AT* is the average exposure time for non-carcinogens (365 days × 70 years) [35].

#### 2.3.7. Hazard Index (HI)

The HI represents the sum of the individual target hazard quotients for each element:(11)$$HI=THQ\left(Cr\right)+THQ(Zn)$$

In the present study, the HI assumes that the consumption of lettuce or onion irrigated with E3 will result in simultaneous exposure to chromium and zinc. If HI > 1, there is the potential for adverse non-carcinogenic health effects.

#### 2.3.8. Statistical Analysis

The data were analyzed using STATISTICA 12 software. The obtained results were presented as the mean of three measurements (the samples from three pots, irrigated with tap water or effluents) ± SD (standard deviation). The Kruskal–Wallis test (*p* < 0.05) was applied to evaluate the significant difference between the chromium and zinc concentrations in the roots and leaves (edible part) of leafy vegetables, as well as in plants and soil, irrigated with industrial wastewater and tap water.

## 3. Results and Discussion

### 3.1. Chromium and Zinc Concentrations in Effluents and Soils

The concentrations of chromium and zinc in three types of industrial effluents and tap water (control), as well as in the corresponding soils, are represented in Table 2.

The zinc concentrations in the industrial effluents E1, E3 and E2 were 1.5, 3.4 and 39 times, respectively, higher than the threshold limits for the element in irrigation waters proposed by the WHO (World Health Organization). The concentration of chromium in the industrial effluent E3 was 100 times higher than established limits (Table 2).

Typically, the accumulation of metals in vegetables has been studied when irrigated with treated wastewater [21,22,38]. However, crop irrigation with untreated industrial and domestic wastewaters was also reported [24,25] (Table 3). According to the literature data, the concentrations of zinc and chromium in the treated effluents used for irrigation varied from 0.009 mg/L to 0.27 mg/L and from 0.003 mg/L to 0.2 mg/L, respectively (Table 3). Data on the concentrations of zinc and chromium in untreated wastewater are very scarce. Souri et al. [25] reported that the concentration of elements in untreated domestic and industrial wastewater ranges from 6.7 mg/L for chromium to 13.1 mg/L for zinc. This exceeds the established standards for chromium and zinc in wastewater used for irrigation by 67 and 6.6 times, respectively [36].

The pH values of the effluents (Table 3) varied within the acceptable range of values (6.5–8) established by the WHO [36], with the exception of a few studies conducted in India and China [3,38]. It is known that irrigation water with a pH outside the normal range may cause nutrient imbalances or contain toxic ions [39]. For example, zinc can be toxic to many plants at a pH < 6.0 [36].

**Table 3 foods-13-03420-t003:** Chromium and zinc accumulation (average ± SD) in the roots and leaves of lettuce and onion irrigated with wastewater across the world.

Study Area	Wastewater Type, pH and Content of Metals (mg/L)	Plants	Plant Parts	Metal Content in Vegetables (mg/kg Dry Weight)		References
				Cr	Zn	
Russia	Industrial effluents E1: Zn—2.95 E2: Zn—78 E3: Zn—6.85; Cr^6+^—9.96	Lettuce	L ^1^		159 ± 7.32 (E1)	Present study
			R ^2^		205 ± 5.56 (E1)	
			L		833 ± 131 (E2)	
			R		1775 ± 431 (E2)	
			L	7.58 ± 2.87 (E3)	133 ± 9.93 (E3)	
			R	200 ± 1.13 (E3)	209 ± 3.67 (E3)	
		Green onion	L		89.5 ± 12.8 (E1)	
			R		599 ± 168 (E1)	
			L		445 ± 39 (E2)	
			R		3827 ± 408 (E2)	
			L	7.36 ± 1.86 (E3)	59.8 ± 0.51 (E3)	
			R	147 ± 37.4 (E3)	438 ± 65.5 (E3)	
Greece	Greenhouse experiment Cr^6+^: 0.01–0.25	Onion	L	0.32–0.66	–	[40]
Nigeria	River water polluted by industrial effluents	Onion	L	6.06 ± 1.28	8.33 ± 0.71	[41,42]
Nigeria	Gray wastewater Zn—3.2	Lettuce	L	–	289–596	[43]
			R		439–1421	
		Onion	L	–	356–456	
			R		450–1158	
Ghana	Anthropogenically polluted water bodies	Lettuce	L	6.96–17.2	–	[11]
		Onion	L	5.51–9.26	–	
Egypt	Domestic and industrial effluents	Lettuce	L	–	118–223	[44]
Egypt	Untreated sewage waters pH—7.25;	Lettuce	L R	–	75 120	[24]
Egypt	Treated wastewater pH—7.87; Cr^3+^—0.003; Zn—0.009	Lettuce	L	1.28 ± 0.05	58.3 ± 1.71	[22]
			R	2.46 ± 0.15	52.8 ± 6.26	
		Onion	L	0.98 ± 0.39	39.9 ± 7.75	
			R	1.17 ± 0.71	63.3 ± 7.27	
United Arab Emirates	Tertiary-level treated municipal wastewater Zn: 0.27 Cr: 0.2	Lettuce	L	2.30 ± 0.30	4.20 ± 0.20	[21]
Iran	Domestic and industrial effluents pH: 7.56 Cr: 6.7 Zn: 13.1	Lettuce	L	12.2 ± 1.65	57.9 ± 8.12	[25]
India	Industrial effluents pH: 7.24 Zn: 0.46 Cr: 2.03	Lettuce	L R	19.5 31.8 ± 10.0	22.0 ± 6.00 30.0 ± 2.00	[45]
India	Industrial effluents, treated sewage waters pH: 6.78–8.13 Zn: 0.05–12.6 Cr: 0.005–4.6	Lettuce	L	1.58–61.4	5.73–171	[3]
China	Treated wastewater pH: 7.8–8.2	Lettuce	L	7.50 ± 7.00	62.0 ± 5.00	[38]
China	Well water (previously irrigated with wastewater)	Onion	L	3.20 ± 0.80	70.0 ± 10.0	[46]
China	Previously irrigated with wastewater	Lettuce	L R	18.2 ± 1.21 47.5 ± 6.70	-	[23]
Different regions of the World	Different types of wastewaters	Onion	L	0.78–3.65	8.15–26.6	[47]
Chinese national standard	-	Vegetables	Edible part	0.5 FW^3^	-	[48]
Russian national standard	-	Vegetables	Edible part	-	10.0 FW	[49]

^1^ L—leaves, ^2^ R—roots, ^3^ FW—Fresh weight.

The concentrations of zinc and chromium in the wastewater-irrigated soils were up to 100 times and 11 times, respectively, higher than those in the control soil (Table 2). In cases where the effluent E2 was used for irrigation, the zinc content in the soil exceeded the upper limit of the recommended permissible concentrations for soils by 4–9 times (Table 2).

It is important to note that the content of zinc and chromium in the wastewater-irrigated soils where the lettuce grew was 2–4 times higher compared to the onion pots. This appears to be primarily due to the greater absorptive capacity of the green onion roots compared to lettuce roots, which will be discussed in Section 3.3.

### 3.2. Chromium and Zinc Accumulation in Leafy Vegetables

The content of chromium and zinc in the leaves and roots of lettuce and onion irrigated with industrial effluents in the present study, as well as the literature data, are summarized in Table 3.

#### 3.2.1. Chromium

The content of chromium in the leaves of onion and lettuce turned out to be approximately the same (7.50 ± 0.16 mg/kg). However, its content in the lettuce roots reached 200 mg/kg and was 1.4 times higher than in the onion roots. The content of chromium in the wastewater-irrigated leaves and roots of leafy vegetables was significantly higher (Kruskal–Wallis test, *p* < 0.04) than in the control plants (Table 3).

Leafy vegetables are known to contain chromium in the range of 0.04–2.4 mg/kg dry weight [37,50], which is several times lower than the content obtained in the present study. Nevertheless, when recalculating based on the fresh weight, the content of chromium in the wastewater-irrigated lettuce (0.40 mg/kg) and green onion (0.036 mg/kg) was lower than the limit of 0.50 mg/kg in fresh vegetables established by China [48]. The WHO does not directly provide a maximum permissible level for chromium in food.

The obtained results are in agreement with the published data on the values of the chromium content in onion and lettuce leaves, which varied from 0.32 mg/kg to 9.26 mg/kg and from 0.98 mg/kg to 19.5 mg/kg, respectively (Table 3). The exception were studies conducted in India and China, where the concentration of chromium in the lettuce leaves was several times or an order of magnitude higher than the present results [3,23,45]. The literature data on the chromium content in the roots of leafy vegetables are scarce. Based on the published data, the content of chromium varied from 1.17 mg/kg in the onion roots to 47.5 mg/kg in the lettuce roots. That is two orders of magnitude or several times, respectively, less than the values obtained in the present study (Table 3).

#### 3.2.2. Zinc

The content of zinc determined in the wastewater-irrigated leaves of lettuce and onion varied widely from 133 mg/kg to 833 mg/kg and from 59.8 mg/kg to 445 mg/kg, respectively, depending on the type of effluent used for watering (Table 3). The roots of the lettuce and onion contained up to 1775 mg/kg and 3827 mg/kg of zinc, respectively. The content of zinc in the wastewater-irrigated leaves and roots of lettuce and onion was significantly higher (Kruskal–Wallis test, *p* < 0.007) than in the control plants.

Leafy vegetables typically contain zinc in the range of 13–334 mg/kg dry weight [37], which is consistent with the data obtained in this experiment for the onion and lettuce irrigated with E1 and E3. Also, the content of zinc in the lettuce (7.18−8.59 mg/kg fresh weight) and green onion (2.03–4.39 mg/kg fresh weight) irrigated with E1 and E3 was lower than the limit of 10 mg/kg in fresh vegetables established in Russia. However, the content of zinc in the leaves of lettuce and green onion irrigated with E2 was 5 and 2 times, respectively, higher than the established standard (Table 3).

The zinc concentrations reported in the literature varied widely, both in leaves and roots, and were in a good agreement with the present data (Table 3). Thus, the zinc content ranged from 4.20 mg/kg and 8.15 mg/kg in the leaves of lettuce and onion, respectively, irrigated with treated wastewater [21] to 596 mg/kg and 356 mg/kg in the leaves of lettuce and onion, respectively, irrigated with gray wastewaters [43]. The data on the zinc content in lettuce and onion roots are very scarce, and the available values vary in s similar range from 30 mg/kg to 1421 mg/kg (Table 3).

#### 3.2.3. Chromium and Zinc Interaction

It was found that the contents of zinc in the leaves of vegetables irrigated with E3, containing both zinc and chromium, were significantly (Kruskal–Wallis test, *p* < 0.007) lower than those in the leaves of the plants irrigated with E1, containing only zinc. Considering that the zinc content in E3 was 2.3 times higher than in E1, an antagonistic effect between zinc and chromium was likely observed.

It is known that the accumulation of metals in vegetables depends on many factors. The presence of a particular metal can also affect the availability of other metals in the soil and therefore in plants due to their antagonistic or synergistic behavior [50,51,52]. The greatest number of antagonistic reactions was observed for Fe, Mn, Cu and Zn, which are essential elements for plant growth. At the same time, it is known that such trace elements as Cr, Mo and Se often enter into antagonistic processes with them [50]. Excessive concentrations of chromium in soil have been reported to interfere with the complete absorption of essential nutrients by plants. Both chromium (III) and chromium (VI) have been found to interfere with the absorption of macronutrients such as N, P, K and Mg [53]. It was also previously reported that chromium inhibits not only photosynthetic and respiration processes in plants but also the intake of essential microelements [15]. Thus, chromium (VI) has been demonstrated to reduce the uptake of Cu, Fe and Zn in *A. viridis* [53]. The inhibition of the translocation of P, K, Zn, Cu and Fe within bean plant parts by exposure to chromium in nutrient solutions has also been described [54]. The effect of chromium on mineral nutrition largely depended on the properties of the soil. Thus, in non-calcareous soils enriched with chromium (III), the translocation of Fe, Zn and Mo to bean plants has decreased [54]. However, other authors [20] who studied the accumulation of various concentrations of chromium in duckweed (*Lemna minor*) in the presence of copper and zinc established a synergistic interaction between chromium and zinc ions in all the combinations of experiments.

### 3.3. Bioconcentration and Translocation Factors

For both plants, the BCF and TF of chromium and zinc were calculated (Table 4). Lettuce and onion actively accumulated zinc (BCF > 1) from the soils with its low content (when irrigated with E1 or tap water). For the soils with high concentrations of zinc (when irrigated with E2 and E3), the opposite pattern was observed (BCF < 1). The values of the BCF of chromium for the leafy vegetables were two orders of magnitude lower than unity when irrigated with both tap water and the effluents.

The TFs of chromium and zinc for the lettuce and onion irrigated with all the types of effluents (except for the TF of zinc for lettuce irrigated with tap water) were less than unity. The content of zinc and chromium in the onion and lettuce leaves, as determined in the present study, was significantly lower (Kruskal–Wallis test, *p* < 0.007 to *p* < 0.02) than in the roots. When the plants were irrigated with the effluents, the TFs of chromium were similar for both plants, and the TFs of zinc for lettuce were 4–5 times higher compared to green onion. Figure 1 and Figure 2 clearly demonstrate the translocation of zinc and chromium in the roots and edible part of leafy vegetables. Thus, in the case of onion, about 90% of the zinc was accumulated in the roots, while in lettuce, zinc was more evenly distributed in the roots and leaves. The percentage of chromium in the edible part of the control plants was several times higher compared to the plants irrigated with the effluents. Thus, the edible part of the control sample of onion contained 32% of chromium, while the edible part of the plant irrigated with the E3 contained only 4% (Figure 2).

The accumulation of metals in plants depends on many factors, including the pH and redox potential of soil, plant species, toxicity threshold of accumulating tissue, and environmental factors such as the proximity of growth to industrial zones or areas of agricultural activity, and others [55,56,57,58]. Among the vegetables, the leafy ones demonstrate the highest BCF due to the higher translocation and transpiration rates [6,59,60]. At the same time, such morphological features of plant organs as the length and biomass of roots, the number of leaves and their area have a huge impact on the intensity of zinc and chromium absorption and accumulation by plants [55]. Thus, longer and thinner green onion roots, characterized by a larger total surface area compared to the roots of lettuce, resulted in more active accumulation of zinc from the soil. In our study, the following pattern was observed: onion roots accumulated 2–3 times more zinc than lettuce roots, while the opposite pattern was found for the zinc concentration in the soil (Table 2). A similar conclusion was reached by the authors of [61], who observed that a thinner root system increased the accessibility of zinc. Perhaps this pattern is typical for only essential elements, since the chromium content in the onion roots was, on the contrary, two times lower than in lettuce roots (Table 3). Similar results were obtained by other researchers [22] who studied chromium accumulation in green onion and lettuce leaves and roots.

Zinc deficiency in crops is much more common than its excess. However, zinc toxicity occurs in contaminated soils, including agricultural soils irrigated with wastewater and treated with sewage sludge [62]. Susceptibility to zinc toxicity largely depends on the type of crop. It is noted that leafy vegetable crops, especially spinach and beets, are sensitive to zinc toxicity [63].

Although the zinc uptake varies depending on the plant species, its absorption is mainly determined by the composition and concentration of the growth media [64]. Plant roots absorb zinc primarily in the form of hydrated zinc, as well as free Zn^2+^ ions, and then it is distributed in a complex manner within the plant [55,65]. Low-molecular-weight complexes, storage metalloproteins, free ions and insoluble structures associated with cell walls are the main forms of zinc. At the intracellular level, zinc is inactivated in all complexes with organic ligands or by complexation with phosphorus [62,64]. It is known that zinc, being a biophilic element, actively accumulates and moves to aboveground plant organs due to its bioavailability in the soil [66]. Zinc has intermediate mobility compared to highly mobile elements, such as K or P, and the immobile Ca [64]. In the laboratory experiment with different types of vegetables and soils, which were subject to frequent treatment with sewage sludge, zinc was one of the most available metals, with the BCFs varying from 1 to 10 [60]. However, in general, the zinc levels in roots were higher than in leaves and shoots [55], which is consistent with the data obtained in this study (TFs < 1).

Plants, while having the ability to absorb metals through their roots, are able to maintain relatively low concentrations of potentially toxic metals in their shoots, avoiding their excessive uptake and transport [56,67,68]. Although some authors reported the stimulating effect of chromium on plants, it is known that chromium is not only a nonessential element for plants but can also be phytotoxic [15,50]. Chromium (III) is poorly available to plants and does not move easily within plants [15,19,50]. The metal is passively absorbed by diffusion at the cation-exchange sites of the cell wall [53]. Chromium (VI) is more available to plants, since it shows higher solubility and mobility in soil compared to chromium (III), but it is the very unstable form under normal soil conditions [57]. It is transported actively along with other critical components, such as sulphate, via sulphate transporters [15,57]. In addition to sulfate transporters, many families of metal transporter genes play critical roles in transporting various metals from roots to shoots, but this has not yet been fully explored for chromium in plants [19]. Chromium has also been observed to compete with sulfur, phosphorus, and iron for carrier binding during transportation [19]. The ability to form complexes with organic acids and mycorrhizal fungi has been shown to increase chromium (VI) uptake by plants [53].

Most researchers have shown that chromium binding occurs predominantly in plant roots, which is mainly caused by its immobilization in the vacuoles of root cells [19,57]. The formation of insoluble chromium compounds inside plants is probably associated with its increased sequestration in plant roots [15]. Chromium is considered to be one of the least transportable elements among the heavy metals in plant roots. The transfer and accumulation of chromium in the soil–plant system also differs depending on the type of crop. Thus, it has been reported that hyperaccumulators of iron (such as spinach) are more effective in translocating chromium to aboveground parts compared to lettuce, which does not accumulates iron [53]. The transfer factor of chromium varies widely from 0.005 to 0.027 [50,57], which is consistent with the TFs determined in this study.

### 3.4. Biochemical Analysis of Leafy Vegetables

In the lettuce samples treated with wastewater containing zinc at a concentration of 2.95 mg/L (E1), the total chlorophyll content, chlorophyll a and chlorophyll b increased by 30–31% compared to the control samples. In the lettuce samples irrigated with wastewater containing zinc at a concentration of 78 mg/L (E2), the total chlorophyll content, chlorophyll a and b were reduced by 34%, 35% and 33% (*p* < 0.05), respectively (Table 5). The total chlorophyll content, as well as the chlorophyll a and chlorophyll b, in the samples irrigated with E3, containing both zinc and chromium, increased by 8.4%, 11% and 4.1%, respectively, compared to the control samples.

Similar changes in the content of total carotene in the samples were observed. In the lettuce samples irrigated with E2, the carotene content decreased by 32% compared to the control, while in the samples treated with E1 and E3, an increase in the carotene content of 37% and 16%, respectively, was observed.

In the lettuce leaves irrigated with E1, there was an increase (*p* < 0.05) in the ability to reduce the ABTS radical by 61%. The DPPH test in these samples showed a two-fold (*p* < 0.05) increase in the antioxidant activity. The content of phenols increased by 70% (*p* < 0.05). The content of phenols and the antioxidant activity of the hydro-ethanolic extracts obtained from the lettuce leaves irrigated with E2 and E3 changed by less than 5% compared to the control samples (Table 5).

In the samples of onion leaves treated with wastewater containing zinc at a concentration of 78 mg/L (E2), the total chlorophyll content increased by 13.5% due to chlorophyll b. Its value was 98% (*p* < 0.05) higher than the control value (Table 6). The content of chlorophyll a in the treated sample was 32% lower compared with the control. The total chlorophyll content was reduced by 39% in the samples irrigated with wastewater containing both zinc and chromium (E3). Low values of chlorophyll a (37% lower than in the control) and chlorophyll b (41% lower than in the control) were determined in these samples. In the onion samples treated with wastewater containing zinc at a concentration of 2.95 mg/L (E1), the chlorophyll content did not change compared to the control. Similar changes were determined in the content of total carotene in the samples. In the samples treated with E2, the carotene content increased by 57% (*p* < 0.05) compared to the control. In the samples irrigated with E3, the carotene content decreased by 29% and the carotene content did not change compared to the control in the samples irrigated with E1 (Table 6).

For the hydro-ethanolic extracts obtained from the onion leaves irrigated with E1 and E3, the ABTS and DPPH test values increased from 4% to 12% compared to the control samples. Conversely, in the onion samples irrigated with E2, the ABTS and DPPH test values were reduced by 11% (*p* < 0.05) and by 8.6% (*p* < 0.05), respectively. The content of phenols increased by 6.6% (*p* < 0.05) in the samples irrigated with E2 and decreased by up to 9.3% (*p* < 0.01) in the samples irrigated with E1 and E3.

Data on the changes in the biochemical composition and antioxidant activity of leafy vegetables when irrigated with wastewater are scarce. The significant decrease in the chlorophyll content observed in the lettuce leaves irrigated with E2 may be due to the inhibition of chlorophyll biosynthesis as a result of the toxic effect of high concentrations of zinc (78 mg/L) on lettuce leaves, as the photosynthetic apparatus is one of the targets of heavy metals in plants [52,69]. Zinc toxicity is largely due to the replacement of other weakly bound divalent metal ions from essential sites, one of which is the Mg^2+^ in chlorophyll. This Mg substitution is known to inhibit photosynthesis [62].

It is well known that metals, depending on their concentration and the plant species they interact with, generate oxidative stress of varying intensities. This state of oxidative stress, in turn, interferes with nearly all the metabolic pathways, including those responsible for the synthesis of photosynthetic pigments [70]. The effect is individual and can manifest in either an increase or decrease in pigment levels. This is evident from our results, where we observed both an increase and a decrease in the chlorophyll and carotenoid content, depending on the metal concentration and plant species involved.

The significantly increased antioxidant activity in the lettuce leaves irrigated with wastewater (E1) may be a protective mechanism induced in response to heavy metal stress. This conclusion was reached by the authors of [69], who assessed the effect of the irrigation of different crops (including lettuce) with industrial wastewater (with zinc content varying from 3 to 6 mg/L) on the growth, photosynthetic pigment content and the antioxidant system. The authors observed a significant increase in the activities of antioxidant enzymes in the wastewater-irrigated plants. It is also likely that an increase in antioxidant activity can be observed when plants are exposed to relatively low concentrations of metals, which can be up to several times higher than the permissible level in water. When plants were irrigated with E2, containing zinc at a concentration that was approximately 40 times higher than the maximum permissible concentration in water for irrigation, no changes in antioxidant activity were observed for lettuce leaves, and for onion leaves, it decreased significantly.

### 3.5. Estimated Daily Metal Intake and Target Hazard Quotients

The EDI and THQ for chromium and zinc estimated through the consumption of lettuce and green onion irrigated with different types of industrial effluents are shown in Table 7. It was observed that no individual THQ for any of the leafy vegetables and elements analyzed is >1 (Table 7). This indicates that in and of themselves, the consumption of lettuce and onion analyzed presents no non-carcinogenic health risk. The EDI and THQ values for chromium estimated for leafy vegetables were similar, while for zinc, the values obtained for lettuce were two times higher than those for onions. This is due to the above-mentioned features of the distribution of elements between the roots and the edible part of leafy vegetables, when lettuce leaves accumulate up to 44% of zinc and onion leaves only up to 13%. The highest THQs were determined for lettuce irrigated with E2 (0.09), containing a high concentration of zinc, and for both vegetables irrigated with E3, containing zinc and chromium (0.08 and 0.07, respectively). The obtained values of the EDI were one–two orders of magnitude lower than the safe daily dietary intake established for zinc and several times lower than those reported for chromium (Table 7). The HI, which considers the cumulative effect of the consumption of chromium and zinc, also did not exceed one. The average percentage contribution of chromium to the hazard index was determined to be 88%.

It should be noted that although Effluent 3 contained hexavalent chromium, the EDI was compared to the data on the safe daily dietary intake for trivalent chromium (Table 7). This is because hexavalent chromium in soil tends to be reduced to trivalent chromium by organic matter. It is also known that readily available Cr (VI) is transformed into the form of Cr (III) in plant cells. In addition, when hexavalent chromium is ingested, it is effectively reduced to its trivalent form in the gastrointestinal tract [50,57].

## 4. Conclusions

This study showed that the accumulation of metals in vegetables irrigated with industrial effluents containing toxic chromium (VI) and zinc in high concentrations followed the following pattern: soil ≥ roots > leaves for lettuce and roots > soil ≥ leaves for green onion. The calculated bioconcentration and translocation factors for both elements turned out to be below unity, and for zinc they decreased with the increase in its concentration in the soil. Thus, although long-term irrigation with wastewater will increase the concentration of zinc in the soil, it will not be proportionally transferred through the food chain.

The influence of high concentrations of zinc was reflected in a number of changes in the biochemical parameters of leafy vegetables. Thus, irrigation with wastewater containing zinc at a concentration of 78 mg/L led to a significant decrease in the content of total chlorophyll, chlorophylls a and b in lettuce and a significant decrease in the antioxidant activity in onion.

The target hazard quotients and hazard indices were less than unity, indicating no non-carcinogenic health risk from the intake of the studied vegetables, primarily due to the fact that a smaller proportion of the total metal content was transferred to the edible part of these leafy vegetables. However, the content of zinc and chromium in the leaves of lettuce and onions, as well as in the soil irrigated with wastewater with the highest zinc concentration (78 mg/L), up to several times–one order of magnitude exceeded the recommended maximum permissible concentrations.

Based on the results obtained, an urgent task for future research is to study the effect of single- and multicomponent industrial effluents with a high concentration of metals on various types of leafy vegetables. It is especially important to identify those species that have a high ability to transfer metals from the roots to edible parts. In addition, since it is known that some elements (for example, chromium) compete with essential elements for binding transporters in plants, an important task will be to study the effect of high concentrations of toxic elements (for example, chromium) on the accumulation and transfer of macro and trace elements in leafy vegetables.

## Figures and Tables

**Figure 1 foods-13-03420-f001:**
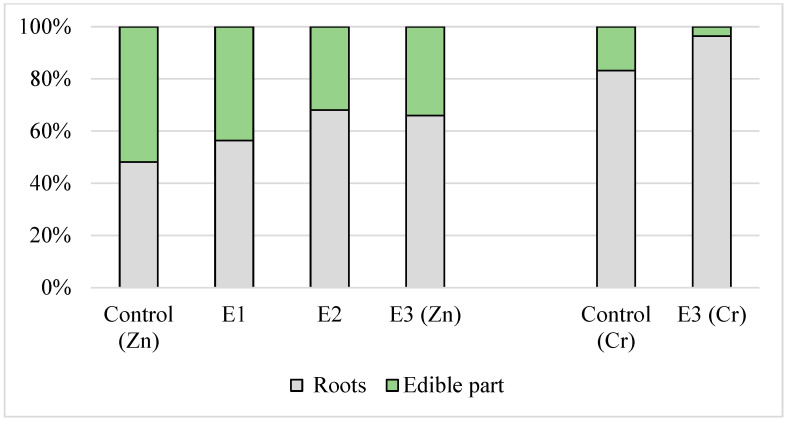
Zn and Cr translocation in the edible part and roots of lettuce irrigated with tap water and industrial effluents.

**Figure 2 foods-13-03420-f002:**
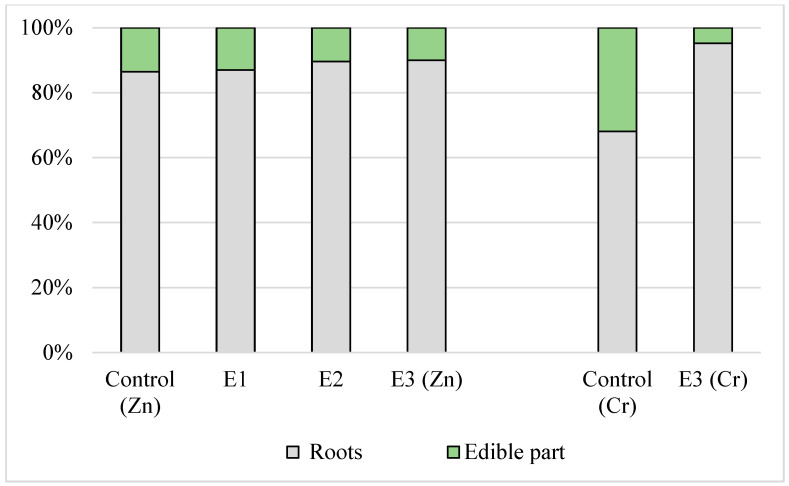
Zinc and chromium translocation in the edible part and roots of onion irrigated with tap water and industrial effluents.

**Table 1 foods-13-03420-t001:** Quality control of the ICP-OES used in the present study.

Element	SRM	Measured Values, mg/kg	Certified Values, mg/kg	Recoveries, %
Cr	OBTL-5	5.27 ± 0.05	6.3 *	84
	2709a	128 ± 0.3	130 ± 9	98
Zn	OBTL-5	53.2 ± 0.34	52.4 ± 3.4	102
	2709a	108 ± 0.6	103 ± 4	105

* Information value for INCT-OBTL-5.

**Table 2 foods-13-03420-t002:** Chromium and zinc concentrations in industrial effluents and the corresponding soils irrigated with wastewater.

Metal	Type of Irrigation Water	Water (mg/L)	Soil (mg/kg)	
			Lettuce	Onion
Zn	Control (tap water)	0.17 ± 0.001	27.5 ± 2.09	22.6 ± 0.58
	E1	2.95 ± 0.01	139 ± 3.33	37.8 ± 0.22
	E2	78.0 ± 0.15	2634 ± 551	1327 ± 274
	E3	6.85 ± 0.04	193 ± 115	90.6 ± 24.1
	International limits	2.0 ^1^	150–300 ^2^	
Cr^6+^	Control (tap water)	<0.0001	18.9 ± 2.31	12.7 ± 0.28
	E3	9.96 ± 0.09	206 ± 98.2	94.4 ± 1.45
	International limits	0.10 ^1^	30–200 ^2^	

^1^ [36], ^2^ [37].

**Table 4 foods-13-03420-t004:** The values of the BCF and TF of chromium and zinc in the lettuce and green onion irrigated with tap water and industrial effluents.

Element	Type of Irrigation Water	C_w_ ^1^ (mg/L)	BCF		TF	
			Lettuce	Onion	Lettuce	Onion
Cr	Tap (Control)	<0.0001	0.03	0.04	0.20	0.47
	E3	9.96	0.04	0.08	0.04	0.05
Zn	Tap (Control)	0.17	2.42	1.65	1.08	0.16
	E1	2.95	1.14	2.37	0.78	0.15
	E2	78.0	0.32	0.34	0.47	0.12
	E3	6.85	0.69	0.66	0.64	0.14

^1^ C_w_—concentration of element in irrigation water.

**Table 5 foods-13-03420-t005:** Biochemical analysis of both wastewater-irrigated and control lettuce leaves.

Irrigation Water	Chlorophyll a	Chlorophyll b	Chlorophyll a + b	Carotene	Phenol	ABTS	DPPH
	mg/100 g					% of inhibition	
Control	46.2 ± 3.88	22.6 ± 4.46	68.8 ± 4.02	11.6 ± 1.65	43.9 ± 14.0	34.0 ± 4.79	23.0 ± 5.59
E1 (Zn)	60.5 ± 14.6	29.4 ± 7.20	89.9 ± 21.7	15.9 ± 5.90	**74.4 ± 18.4**	**54.8 ± 13.5**	**47.3 ± 15.7**
E2 (Zn)	**30.2 ± 3.36**	**15.1 ± 2.56**	**45.3 ± 5.74**	7.88 ± 1.50	41.5 ± 8.63	33.7 ± 8.06	22.7 ± 9.40
E3 (Cr + Zn)	51.1 ± 6.62	23.5 ± 6.14	74.6 ± 2.78	13.5 ± 2.99	42.4 ± 12.8	34.9 ± 3.55	24.0 ± 4.14

In bold—a significant difference between the experimental samples and the control ones was observed (*p* < 0.05).

**Table 6 foods-13-03420-t006:** Biochemical analysis of both wastewater-irrigated and control onion leaves.

Irrigation Water	Chlorophyll a	Chlorophyll b	Chlorophyll a + b	Carotene	Phenol	ABTS	DPPH
	mg/100 g					% of inhibition	
Control	52.9 ± 8.49	28.6 ± 6.50	81.5 ± 15.0	13.4 ± 2.61	170 ± 1.06	52.9 ± 1.01	43.4 ± 0.35
E1 (Zn)	52.7 ± 14.0	28.7 ± 7.59	81.4 ± 21.6	13.2 ± 3.27	159 ± 6.36	56.1 ± 2.02	47.2 ± 2.12
E2 (Zn)	36.0 ± 3.62	**56.5 ± 3.01**	92.5 ± 6.64	**21.0 ± 0.93**	**181 ± 1.41**	**46.9 ± 0.10**	**39.7 ± 0.47**
E3 (Cr + Zn)	37.7 ± 1.83	20.5 ± 0.21	58.2 ± 1.63	9.47 ± 0.03	**154 ± 0.71**	54.9 ± 0.91	**48.7 ± 0.24**

In bold—a significant difference between the experimental samples and the control was observed (*p* < 0.05).

**Table 7 foods-13-03420-t007:** Estimated daily intake, target hazard quotient and total hazard index of chromium and zinc via the consumption of lettuce and onion irrigated with industrial effluents.

		EDI (µg/kg bw day)	THQ	HI
Lettuce	E1 (Zn)	4.89	0.02	-
	E2 (Zn)	25.7	0.09	-
	E3 (Zn)	4.10	0.01	0.09
	E3 (Cr)	0.23	0.08	
Onion	E1 (Zn)	2.50	0.01	-
	E2 (Zn)	12.5	0.04	-
	E3 (Zn)	1.67	0.01	0.08
	E3 (Cr)	0.21	0.07	
International standards	Zn	200–285 ^1^	<1	<1
	Cr	0.71–2.9 ^1^		

^1^ Safe daily dietary intake (food) [71,72].

## Data Availability

The original contributions presented in the study are included in the article, further inquiries can be directed to the corresponding author.

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
