# Peer review of "Impact of Metal-Containing Industrial Effluents on Leafy Vegetables and Associated Human Health Risk"

_foods, 2024, doi:10.3390/foods13213420_

Round 1

Reviewer 1 Report

Comments and Suggestions for Authors

The present manuscript seems fine but there are several concerns regarding some specific data which must be added to understand the study better.

Comments on the Quality of English Language

Overall language is understandable but I would suggest the authors to carefully read the manuscript and edit wherever the meaning is not clear.

Reviewer 2 Report

Comments and Suggestions for Authors

I was tasked to review the manuscript “Impact of metal-containing industrial effluents on leafy vegetables and associated human health risk” for the journal Foods.

The article is well written and investigated an interesting research field often neglected by the scientific literature. No exceeding of self-citation was detected (no one to be honest). My general opinion is positive but there are several weaknesses that must be reinforced before publication. Authors focused all metal measurements on two metals but with the ICP they can quantify many of them in same run. Evaluating more elements mean a much stronger impact of the work. Authors are invited to take a look of their data and to pick up the amount of all metals. In addition, when it comes to deal with environmental contamination also speciation is crucial. Not all oxidation forms are the same, especially for a paper aimed to evaluate risks on human health. In addition, authors based their manuscript on 66 references but most of them are more than 10 years old. I invite the authors to further investigate literature and refer their findings on more recent papers. Finally, considering only 3 wastewater types cannot be considered representative.

Some more specific comments are following:

Lines 57-67. Check the line spacing.

Line 73. It is worth mentioning that several other environmental aspects can impact the concentration of metals, both inside and outside, of plants such as industrialization, proximity to highways and geographical characteristics. This was deeply investigated in a recent study on wheat that I recommend the authors to read and cite (10.3390/molecules29133148).

Line 76-78. This is not what the authors did in the manuscript. To do so a totally different experimental set-up must be designed. In this paper the authors studied the effects on vegetables, what they can do on human health was not investigated. The THQ studies are not sufficient to say so.

Line 79. Check the English form.

Line 79-90. This should be moved to the experimental part.

Line 93: what was the location of the sampled sites? What types of industrial production where they related to?

Line 101-103. Check chemical formulas and formatting of subscript coefficients.

Line 123: what was the digestion procedure?

Line 129: what was the calibration range?

Line 138. What was the instrument used for the measurement of absorbance? Did the authors evaluate also blanks?

Table 1: to have a comprehensive overview, samples of lettuce and onion of the same species irrigated with regulated tap water must be analyzed in the same batch.

Line 280: this is a relevant finding. Authors are invited to further investigate it and to implement a discussion with literature data.

Line 502: this title is misleading to me. Something like “THQ ranking” should look more appropriate to me.

Lines 530-538: this can not be intended as a conclusion.

As reported above, my opinion is positive but in this form this manuscript looks partial and worths a strong improvement from many points of view. Major revisions are requested.

Reviewer 3 Report

Comments and Suggestions for Authors

The manuscript deals with the effects of untreated wastewater, containing high concentrations of Cr and Zn, on lettuce and green onion and the potential human health risks associated with their consumption, using a pot experiment. The topic is interesting, in the context of worldwide contamination with metals and their deleterious effects. The experiments seem properly performed and the results correctly interpreted, but the following comments should be addressed:

Please mention briefly the thermic program used for samples digestion using the microwave system.

Please mention in the manuscript the purity, the manufacturer and the country of origin of the reagents used for analyses, and also the country of origin of ICP-OES. Please indicate the type and the producer of spectrophotometer/instrument use for absorbance measurements.

Please mention the limits of detection/quantitation for Zn and Cr analysed by ICP-OES. Also, please mention the instrumental parameters used for metal analysis. The authors should add in the manuscript the standard solutions (type, concentrations, producer etc.) used for instrument ICP-OES calibration.

The QA/QC is missing. The authors reported the use of two CRMs, but the obtained results are missing.

In the Abstract it was mentioned that “the concentrations of chromium and zinc in the edible parts of vegetables ranged between 7.36 – 7.58 mg/kg dry weight and 59.8 – 833 mg/kg dry weight, respectively”. In the Table 2, the Cr concentrations of 7.36 mg/kg dw and 7.58 mg/kg dw represent averaged values. In Table 2: it should be noted  hat the values represent the mean concentration ± standard deviation.

Just as a personal opinion, it would have been interesting to study the influence of Cr on the absorption of macronutrients such as N, P, K and Mg and microelements (Fe, Cu etc.) in soil and their translocation in plants.

Reviewer 4 Report

Comments and Suggestions for Authors

Dear authors

 I believe this study is out of scope for the target purpose referred in the manuscript, because:

1) is based on a one-pot experiment, and the continuous irrigation with untreated contaminated industrial effluents over time in the same soil should be investigated.

2) The United Nations Food and Agriculture Organisation (FAO) does not view chromium as an essential growth element and recommends a maximum dosage of 0.1 mg L−1 in agricultural water for irrigation. This study applies contaminated water well above this limit (9.96 mg/L).

3) Using continuous irrigation of untreated contaminated industrial effluents causes an issue with the polluting of subsurface water.

Given these three points, it could be presented in a different journal (not food area) with a different purpose.

Comment 1: Table 1 needs to be reorganized. The picked borders make it tough to read and understand. Replace "International standards" with "international limits". In line 268, change "limit" for "standard".

Comment 2: discriminate the chromium valence form studied.

Round 2

Reviewer 2 Report

Comments and Suggestions for Authors

Authors did an excellent job improving the manuscript and strenghtening most lacks highighted during the first revision round. Even though not all the requests have been satisfied the manuscript now is much more robust and is suitable for pubblication.

Reviewer 3 Report

Comments and Suggestions for Authors The manuscript has been sufficiently improved to warrant publication in Foods.